# Association between Otosclerosis and Rubella in Taiwan: A Nationwide Case-Control Study

**DOI:** 10.3390/jcm12051761

**Published:** 2023-02-22

**Authors:** Juen-Haur Hwang, Ben-Hui Yu, Yi-Chun Chen

**Affiliations:** 1Department of Otolaryngology Head and Neck Surgery, Dalin Tzu Chi Hospital, Buddhist Tzu Chi Medical Foundation, Chiayi 622, Taiwan; 2School of Medicine, Tzu Chi University, Hualien 970, Taiwan; 3Department of Medical Research, China Medical University Hospital, China Medical University, Taichung 404, Taiwan; 4Department of Radiation Oncology, Dalin Tzu Chi Hospital, Buddhist Tzu Chi Medical Foundation, Chiayi 622, Taiwan; 5Division of Nephrology, Department of Internal Medicine, Dalin Tzu Chi Hospital, Buddhist Tzu Chi Medical Foundation, Chiayi 622, Taiwan

**Keywords:** case-control study, otosclerosis, rubella, virus

## Abstract

Otosclerosis is an early adult-onset disease that is associated with 5–9% and 18–22% of all cases of hearing and conductive hearing loss, respectively, and it is suspected to have a viral etiology. However, the role of viral infection in otosclerosis is still inconclusive. This study aimed to investigate whether rubella infection was associated with otosclerosis risk. We conducted a nationwide case-control study in Taiwan. Data were retrospectively analyzed from the Taiwan National health Insurance Research Database. Cases consisted of all patients who were aged ≥6 years and had a first-time diagnosis of otosclerosis for the period between 2001 and 2012. The controls were exact matched to cases in a 4:1 ratio by birth year, sex, and must survive in the index year of their matched cases. Adjusted odds ratio (OR) and 95% confidence interval (CI) were estimated by using conditional logistic regression. We examined 647 otosclerosis cases and 2588 controls without otosclerosis. Among the 647 patients with otosclerosis, 241 (37.2%) were male and 406 (62.8%) were female, with most aged between 40 and 59 years, with a mean age of 44.9 years. After adjusting for age and sex, conditional logistic regression revealed that exposure to rubella was not associated with a significant increase in otosclerosis risk (adjusted OR, 2.0; 95% CI, 0.18–22.06, *p* = 0.57). In conclusion, this study did not show that rubella infection was associated with the risk of otosclerosis in Taiwan.

## 1. Introduction

Otosclerosis is a disorder that results in adult-onset conductive and/or mixed type hearing loss. It is associated with 5–9% and 18–22% of all cases of hearing loss and conductive hearing loss, respectively. The prevalence of otosclerosis is 0.3–0.4% in Caucasians [1]. The prevalence of clinical otosclerosis was approximately 0.1% in Lithuania in 1975. Those aged between 16 and 50 years are at risk of clinical otosclerosis, with the greatest risk being in those aged 20–40 years [2]. In Tunisia, the prevalence of otosclerosis varies between 0.4% and 0.8%, and the incidence of otosclerosis is high in the 26–35 year age group [3]. The age of onset and incidence of bilateral disease, tinnitus, and vertigo, were higher in women (65%) than in men (35%) in a British cohort [4]. Marinelli et al. [5] reported that the incidence of otosclerosis rose from the 1950s (8.9 per 100,000 person-years) to a peak of 18.5 per 100,000 person-years) between 1970 and 1974. Thereafter, the incidence gradually declined to 3.2 per 100,000 person-years between 2015 and 2017.

Otosclerosis is typically characterized by labyrinthine endochondral sclerosis and loss of the free motion of the stapes. Some environmental factors are associated with otosclerosis, such as estrogen exposure, fluoride insufficiency, disturbed bone metabolism, persistent measles virus infection, and autoimmunity [6,7,8]. However, a case-control study reported that there was no relationship between endogenous estrogen exposure and the development of otosclerosis. Women with otosclerosis who had a history of pregnancy did not have significantly worse hearing at the time of diagnosis [9]. Moreover, previous studies found that genetic factors might play a significant role in the manifestation of otosclerosis [6,7,8].

The viral etiology of otosclerosis suggests that otosclerosis is an inflammatory reaction of the otic capsule initiated or caused by viruses such as measles, rubella, and mumps viruses [10,11,12]. However, Singh et al. [13] reported that otosclerosis is not commonly associated with measles, rubella, human cytomegalovirus, herpes simplex, varicella zoster, and Ebstein–Barr virus infections. Thus, the role of viral infection in otosclerosis is still inconclusive based on tissue immunohistochemical, molecular or serological testing in very limited cases. Thus, this study aimed to investigate the relationship of rubella infection and otosclerosis based on a nationwide, population-based case-control study.

## 2. Materials and Methods

### 2.1. Data Source

In this population-based case-control study, data to be analyzed were obtained from the Taiwan National Health Insurance (NHI) Research Database (NHIRD) between 2001 and 2012. The NHIRD is retrieved from a random sample of all beneficiaries in the Taiwan NHI program and there is no significant difference in the sex distribution, age distribution, or average insured payroll-related amount between the patients in the NHIRD and the NHI program. The NHI program, launched in 1995, is compulsory for all residents of Taiwan and covers almost all healthcare services, with a coverage rate of approximately 99%. Thus, the NHIRD has comprehensive healthcare information for research purposes except for laboratory and lifestyle data and adopts International Classification of Diseases-Ninth Revision-Clinical Modification (ICD-9-CM) diagnostic codes to define diseases.

The NHIRD is a deidentified database, in which all personal information is anonymous, and it has been described in detail in our previous studies [14,15,16,17]. Thus, this study was exempt from full review by The Research Ethics Committee of Dalin Tzu Chi Hospital, Buddhist Tzu Chi Medical Foundation (No. B10202022). The requirement for informed consent was waived because the study was a retrospective data analysis.

### 2.2. Identification of Cases and Controls

Cases consisted of all patients who were aged ≥6 years and had a first-time diagnosis of otosclerosis (ICD-9-CM codes 387.0, 387.1, 387.2, 387.8 and 387.9), which is characterized by progressive conductive hearing loss, a normal tympanic membrane, and no evidence of middle ear inflammation [18], between 1 January 2001, and 31 December 2012. The date of the first diagnosis of otosclerosis was defined as the index date for each case. The remaining patients without otosclerosis were used as controls. We performed an exact match by birth year and sex, and randomly selected four controls per case. The controls must survive in the index year of their matched cases, and they were without any otosclerosis diagnosis. For controls, the index date was within the same year of the index date (date of first-time diagnosis of otosclerosis) of their matched cases.

### 2.3. Exposure to Rubella

Exposure to rubella was defined as an ICD-9-based diagnosis of rubella (ICD-9-CM code 056) [19]; rubella is characterized by fever, generalized erythematous maculopapular rash, lymphadenopathy and positive anti-rubella IgM and/or IgG [20] at any time between 1 January 2001, and the index date.

### 2.4. Statistical Analysis

We calculated the distribution of otosclerosis cases in each sex, age interval, and study year. Categorical and continuous variables of the otosclerosis and non-otosclerosis groups were compared using the chi-square test and t-test, respectively. Odds ratio (OR) and 95% confidence interval (CI) for otosclerosis in those with and without rubella infection were estimated using conditional logistic regression models stratified by age, sex, and index year. All statistical tests were two-sided, and the significance level was set at 0.05. All data in this study were analyzed using SAS (version 9.4; SAS Institute, Inc., Cary, NC, USA) and SPSS (version 20.0; IBM Corp., Armonk, NY, USA).

## 3. Results

### 3.1. 12-Year Profile of Otosclerosis

Over a 12 year period, a total of 647 otosclerosis cases were identified (Table 1); 241 (37.2%) were male and 406 (62.8%) were female, and the mean age was 44.9 years. Most of the patients were aged 40–59 years. The prevalence of otosclerosis in Taiwanese patients aged ≥6 years between 2001 and 2012 was 0.033%.

### 3.2. Baseline Characteristics

Records from 647 otosclerosis cases and 2588 selected matched controls were included in the analyses. Table 2 presents the distribution of demographic characteristics of the otosclerosis and control groups. No significant difference was noted in sex and age between the cases and controls.

### 3.3. Association between Rubella Infection and Otosclerosis Risk

The relationship between rubella and otosclerosis is shown in Table 3. There were no cases of exposure to measles. Exposure to rubella was not associated with an increased risk of otosclerosis after adjusting for age and sex (adjusted OR = 2.0, 95% CI = 0.18–22.06, *p* = 0.57).

## 4. Discussion

This nationwide, population-based case-control study did not show that rubella infection was associated with the risk of clinical otosclerosis in Taiwan. This large-scale clinical study supported the findings of Singh et al. [13], but against some reports of immunohistochemical studies [10,11,12].

Stapes ankylosis is a heterogeneous disease that causes conductive hearing loss, and it has different causes. Non-otosclerotic stapes fixation may be a degenerative disorder with variable histopathology. However, otosclerosis is a bone-remodeling disorder of the human otic capsule, but its etiopathogenesis remains unclear. Genetic predisposition, impaired bone metabolism, persistent measles infection, autoimmunity, and hormonal and environmental factors may contribute to the pathogenesis of otosclerosis [5,21]. Moreover, a variety of pathways have been identified to be involved in the development of otosclerosis [22].

Approximately 40% of patients with otosclerosis have a positive family history, and such patients have an earlier age of onset and a higher incidence of bilateral disease and vertigo than those without a family history of otosclerosis. Pedigree analysis is consistent with an autosomal dominant inheritance with reduced penetration [4]. Genetic association studies and gene expression analysis of otosclerotic bone showed that the transforming growth factor-beta 1 pathway is an important factor in the pathogenesis of otosclerosis [8]. Linkage analysis of large families with a history of otosclerosis also showed that the T cell receptor beta is a gene responsible for familial otosclerosis [22]. These studies suggest that some immunological pathways are related to familial otosclerosis [8,22].

Otosclerosis is also considered an inflammatory reaction of the otic capsule initiated or caused by viruses such as measles, rubella, or mumps [10,11,12]. Among all viruses, the majority of the current literature highlights the importance of the measles virus component in otosclerotic stapes samples and its role in the etiopathogenesis of otosclerosis [1,6,7,8,23,24,25]. The presence of tumor necrosis factor-α and interleukin-1β mRNA in virus-positive stapes could be the result of viral antigen stimulation [25]. In Germany, between 1993 and 2004, a highly significant decrease in otosclerosis was noted among the population vaccinated against the measles virus. These experimental and clinical correlations indicate that the measles virus is an important trigger of otosclerosis [26].

However, other studies have failed to show the presence of the measles virus component in otosclerotic stapes [1,24]. Although measles virus RNA has been detected in otosclerotic samples using reverse transcriptase-polymerase chain reaction (RT-PCR), no complete measles virus mRNA sequence has been reported and no infectious virus has been isolated from clinical samples in Japan [27]. Moreover, Flores-García et al. [28] reported that no sample was positive for any of three measles virus genes (H, N, and F) from the stapes of 93 patients with otosclerosis. Measles virus RNA was not detected in any sample via real-time RT-PCR. Further, CD46 levels were positive only in 3.3% of otosclerosis cases in a British cohort [4], and Singh et al. [13] reported that otosclerosis is not commonly associated with systemic viral infections, such as measles, rubella, human cytomegalovirus, herpes simplex, varicella zoster, and Epstein–Barr virus by testing their IgM antibodies. Similarly, in our study based on the Taiwan population, we found that rubella infection did not have a positive association with otosclerosis.

The discrepancy between many previous results and/or our study might be due to different study methodologies and/or case numbers. That is, different stages of otosclerosis, different parts of the surgical specimen, different types of immunological or serological antibodies, different sequences of PCR primers, and unsuitable control specimens or participants might result in different or contradictory results. In our study, we selected cases and controls who were aged >6 years from the Taiwan NHIRD between 1997 and 2012. Thus, our study covered a wider age range and more cases than previous studies, considering surgical specimen and serological testing. Moreover, viral infection might act as an initiator of otosclerosis at earlier ages in those with a genetic background, and the virus may or may not be detected when such patients get. Actually, a clinical diagnosis for both rubella infection and otosclerosis would be sufficient while assessing the relationship between the two conditions.

To our knowledge, this is the first study to use a relatively large sample size from universal coverage of a nationwide population, long-term track all exposures to rubella, and undertake an exact match to document no association between otosclerosis risk and rubella infection. However, this study has some limitations. First, given the baseline risk p_0_ = 0.001 and under the conditions of detecting an odds ratio (OR) of two with 80% power at the 5% significance level and that of a control-to-case of four was planned, the minimum number of cases required was estimated to be 15841. If a control-to-case of 40 was planned, the minimum number of cases required was estimated to be 11276. Our otosclerosis cases identified were lower than the estimated sample size. This was attributable to the lower prevalence (0.033% between 2001 and 2012) of otosclerosis in Taiwan. In addition, potentially undiagnosed cases in the controls are a common and real limitation in observational studies. Thus, this bias might reduce the possibility of a positive association between rubella and otosclerosis. Second, the diagnosis of rubella relied on ICD-9-CM code in this study. Rubella has been listed as a mandatory notifiable disease in Taiwan since 1988 [29] and must be reported to Taiwan Centers for Disease Control via ICD-9 code. Thus, the diagnostic accuracy of rubella is believable. Due to high coverage rate with an effective vaccination plan and surveillance system initiated in 1991 [30], the incidence of reported rubella cases has decreased dramatically to fewer than 10 cases per million individuals since 1994 in Taiwan [29]. Rubella has been brought under effective control in the beginning of 1998 and less than 60 confirmed rubella cases were reported annually [30]. Thus, this might account for the scant rubella cases identified in our study between 2001 and 2012. Moreover, diagnoses of exposure to rubella has relied mostly on clinical information and positive IgM. However, it is well known that these are not reliable tools for rubella diagnosis, especially in the vaccination era. Currently rubella diagnosis relies on PCR and/or low IgG avidity (when IgM are positive) [20]. IgM for rubella might be also positive following vaccination and/or other viral rash, and may persist for several years [20]. Besides, no information is provided on vaccination of the patients. Thus, the association between otosclerosis risk and rubella infection was not observed and/or that the vaccine is beneficial in the prevention of otosclerosis in Taiwan. Third, exposure to estrogen could not be assessed in the NHIRD. Fourth, the true interval between exposure to rubella infection and diagnosis of otosclerosis was unknown. In this case-control study, rubella infection was observed before and up to the year of the first diagnoses of otosclerosis. There was only one case of otosclerosis with rubella infection and both diagnoses of otosclerosis and rubella infection were in the same year. With such a small number, it was difficult to make inferences on the delay between rubella infection and otosclerosis. Nevertheless, our findings may be used as a basis for further investigations.

## 5. Conclusions

This nationwide, population-based case-control study did not show that rubella infection was associated with clinical otosclerosis in Taiwan. Furthermore, in the view of preventive medicine, a follow-up study is warranted to determine whether the risk of otosclerosis would decrease further after a vaccination program against rubella.

## Figures and Tables

**Table 1 jcm-12-01761-t001:** Distribution of otosclerosis by sex and age levels in 2001–2012, Taiwan.

	Otosclerosis (n = 647)		*p*-Value
	Male (n = 241) N, %	Female (n = 406) N, %	
Total mean age (±SD)	44.9 ± 18.6		
Age (year, mean ± SD)	44.5 ± 21.1	45.0 ± 17.0	0.74
Age level (year)			
6–19	36 (14.9)	40 (9.9)	
20–39	66 (27.4)	103 (23.4)	
40–59	79 (32.8)	183 (45.1)	
60–79	46 (19.1)	75 (18.5)	
≥80	14 (5.8)	5 (1.2)	

Abbreviations: SD, standard deviation.

**Table 2 jcm-12-01761-t002:** Baseline characteristics between otoscelrosis cases and non-otosclerosis controls after an exact match.

	Otoscelrosis				*p*-Value
	Yes, N = 647		No, N = 2588		
	n	%	n	%	
Sex					1.00
Female	406	62.8	1624	62.8	
Male	241	37.2	964	37.2	
Age level (year)					1.00
6–19	76	11.8	304	11.8	
20–39	169	26.1	676	26.1	
40–59	262	40.5	1048	40.5	
60–79	121	18.7	484	18.7	
≥80	19	2.9	76	2.9	
Mean (SD)	44.9	(18.6)	44.9	(18.6)	1.00

Categorical variables given as number (percentage); continuous variable, as mean ± standard deviation (SD).

**Table 3 jcm-12-01761-t003:** Associations between rubella and otosclerosis risk in 2001–2012, Taiwan.

	No. of Otosclerosis/ No. of Controls		Crude			Adjusted *	
		OR	95% CI	*p*	OR	95% CI	*p*
Before propensity matching (n = 1,983,154)							
No rubella	646/1,981,188	1.00	Reference		1.00	Reference	
Rubella (+)	1/1319	2.34	0.33–16.55	0.40	2.44	0.34–17.35	0.37
After propensity matching (n = 3235)							
No rubella	646/2586	1.00	Reference		1.00	Reference	
Rubella (+)	1/2	2.01	0.18–22.15	0.57	2.00	0.18–22.06	0.57

Abbreviations: CI, confidence interval; OR, odds ratio. * Adjusted for age by year and sex.

## Data Availability

Restrictions apply to the availability of these data. Data was obtained from National Health Insurance database and are available from the authors with the permission of National Health Insurance Administration of Taiwan.

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
