# Peer review of "Association between Otosclerosis and Rubella in Taiwan: A Nationwide Case-Control Study"

_jcm, 2023, doi:10.3390/jcm12051761_

Round 1
Reviewer 1 Report
Main comment
Firstly, exposure to rubella seems to rely on clinical information and positive IgM. However, it is well known that these are not reliable tools for rubella diagnosis (especially in vaccination era). Currently rubella diagnosis relies on PCR and/or low IgG avidity (when IgM are positive).
Secondly, no information is provided on vaccination of the patients. IgM are also positive following vaccination and may persist a very long time (several years).
Finally, did the authors investigate delay between "rubella infection" and otosclerosis onset? Please eventually provide information on this point and discuss
For all these reasons it seems that exposure to rubella virus is not certain (vaccination?) and delay between investigation and rubella also uncertain (IgM may be positive following non specific stimulation of the immune system that may occur following another viral on non-viral rash (parvovirus B19 for example)). This crucial point calls into question the whole concept of this study.
Author Response
- Firstly, exposure to rubella seems to rely on clinical information and positive IgM. However, it is well known that these are not reliable tools for rubella diagnosis (especially in vaccination era). Currently rubella diagnosis relies on PCR and/or low IgG avidity (when IgM are positive).
Response: To address the reviewer’s concern, we revised it in the discussion of the revised manuscript (page 5, lines 202-206).
- Secondly, no information is provided on vaccination of the patients. IgM are also positive following vaccination and may persist a very long time (several years).
Response: To address the reviewer’s concern, we revised it in the discussion of the revised manuscript (page 5, lines 206-207).
- Finally, did the authors investigate delay between "rubella infection" and otosclerosis onset? Please eventually provide information on this point and discuss.
Response: To address the reviewer’s concern, we revised it in the discussion of the revised manuscript (page 6, lines 210-215).
- For all these reasons it seems that exposure to rubella virus is not certain (vaccination?) and delay between investigation and rubella also uncertain (IgM may be positive following non specific stimulation of the immune system that may occur following another viral on non-viral rash (parvovirus B19 for example)). This crucial point calls into question the whole concept of this study.
Response: To address the reviewer’s concern, we revised it in the discussion of the revised manuscript (page 5, lines 202-209; page 6, lines 210-215).

Reviewer 2 Report
Dear Authors
It was a pleasure to read the work on the relationship between rubella virus infection and the incidence of otosclerosis in the Taiwanese population.
The work required analysis of a large population group and resulted in the observation of no association between infection and disease. The paper is well written and presents the results objectively and their limitations. My only reservation is that rubella virus vaccination is available for the Taiwanese population. Therefore, it is possible that the results presented could be different for an unvaccinated population. However, this work may indicate that there is no effect of infection or that the vaccine is beneficial in the prevention of otosclerosis, which should be emphasised in the discussion.
I recommend the work for publication.
Author Response
- However, this work may indicate that there is no effect of infection or that the vaccine is beneficial in the prevention of otosclerosis, which should be emphasized in the discussion.
Response: To address the reviewer’s concern, we revised it in the discussion of the revised manuscript (page 5, lines 208-209).

Reviewer 3 Report
This article considers the potential role of rubella virus infection as a aetiological factor in subsequent development of otosclerosis. The text is mostly well written although some sentences need to be reviewed and rewritten in order to make sense. The research is reliant upon accurate numbers of cases of otosclerosis. What is not alluded to is that there will probably be undiagnosed cases of otosclerosis within the population and unless all the controls were subjected to audiometry there is no way of knowing that none of them have a conductive hearing loss with normal tympanic membranes. With a prevalence of 0.033 there may be 75 cases of undiagnosed otosclerosis in the control sample - has this been considered?
How does this influence your conclusions?
Author Response
- The text is mostly well written although some sentences need to be reviewed and rewritten in order to make sense.
Response: To address the reviewer’s concern, we received an English editing service and attached a certificate of English editing.
- The research is reliant upon accurate numbers of cases of otosclerosis. What is not alluded to is that there will probably be undiagnosed cases of otosclerosis within the population and unless all the controls were subjected to audiometry there is no way of knowing that none of them have a conductive hearing loss with normal tympanic membranes. With a prevalence of 0.033 there may be 75 cases of undiagnosed otosclerosis in the control sample - has this been considered? How does this influence your conclusions?
Response: To address the reviewer’s concern, we revised it in the discussion of the revised manuscript (page 5, lines 195-197).

Round 2
Reviewer 1 Report
Even if the discussion has included my remarks, I still believe that bias in identification of rubella cases call into question the whole concept of this study and the conclusion of authors
Author Response
Feb 5, 2023
George Psillas, Editor in special issue "Hearing Disorders: Diagnosis, Management, and Future Opportunities: Part II "
Journal of Clinical Medicine
Re: jcm-2198883
Dear Editors,
Thanks for your letter on Feb 4, 2023 regarding our article titled as: Association between otosclerosis and rubella in Taiwan: A nationwide case-control study. We've learned a lot from your valuable advice. According to your suggestion, we make some revisions marked as yellow color in the revised manuscript. More clear details are described as the follows.
Response to Reviewer 1
- Even if the discussion has included my remarks, I still believe that bias in identification of rubella cases call into question the whole concept of this study and the conclusion of authors.
Response: The diagnosis of rubella relied on ICD-9-CM code in this study and prior NHIRD-based research [Ref: 19]. Rubella has been listed as a mandatory notifiable disease in Taiwan since 1988 [Ref: 29] and must be reported to Taiwan Centers for Disease Control via ICD-9 code. Thus, the diagnostic accuracy of rubella is believable. Due to high coverage rate with an effective vaccination plan and surveillance system initiated in 1991 [Ref: 30], the incidence of reported rubella cases has decreased dramatically to fewer than 10 cases per million individuals since 1994 in Taiwan [Ref: 29]. Rubella has been brought under effective control in the beginning of 1998 and less than 60 confirmed rubella cases were reported annually [Ref: 30, the following Figure in the revised manuscript]. Thus, this might account for scanty rubella cases identified in our study between 2001 and 2012. To address the reviewer’s concern, we revised our manuscript (page 2, line 95; page 5, lines 197-206) and added three references (Ref 19, 29, 30).
Figure in the revised manuscript: Available at Taiwan Centers for Disease Control [Ref: 30]: https://www.cdc.gov.tw/En/Category/ListContent/bg0g_VU_Ysrgkes_KRUDgQ?uaid=5OqfXlJc1CDMqKnytW3Qww.
Thank you heartily for your invaluable opinions on this paper. We are deeply honored by the time and efforts that you had spent in reviewing and revising this manuscript. By incessantly reviewing and revising our texts, we are spurred to read more and learn more from your comments.
Yours sincerely,
Yi-Chun Chen, MD
Division of Nephrology, Department of Internal Medicine, Buddhist Dalin Tzu Chi General Hospital, Chiayi, and School of Medicine, Tzu Chi University, Hualien, Taiwan
